# Sustainable Valorization of Coffee Silverskin: Extraction of Phenolic Compounds and Proteins for Enzymatic Production of Bioactive Peptides

**DOI:** 10.3390/foods13081230

**Published:** 2024-04-17

**Authors:** Wilasinee Jirarat, Tanyawat Kaewsalud, Kamon Yakul, Pornchai Rachtanapun, Thanongsak Chaiyaso

**Affiliations:** 1Interdisciplinary Program in Biotechnology, Multidisciplinary and Interdisciplinary School, Chiang Mai University, Chiang Mai 50100, Thailand; jirarat4824@gmail.com (W.J.); tanyawant095@gmail.com (T.K.); 2Division of Biotechnology, Faculty of Agro-Industry, Chiang Mai University, Chiang Mai 50100, Thailand; kamon.y@cmu.ac.th; 3Division of Packaging Technology, Faculty of Agro-Industry, Chiang Mai University, Chiang Mai 50100, Thailand; pornchai.r@cmu.ac.th

**Keywords:** phenolic compounds, coffee silverskin valorization, protein hydrolysis, hydrothermal extraction, microwave-assisted alkaline extraction, ultrasound-assisted alkaline extraction, bioactive compounds

## Abstract

Coffee silverskin (CS), a by-product of the coffee roasting process, has high protein content (16.2−19.0%, *w*/*w*), making it a potential source for plant protein and bioactive peptide production. This study aims to develop innovative extraction methods for phenolic compounds and proteins from CS. The conditions for hydrothermal (HT) extraction of phenolic compounds from CS were optimized by varying CS loading (2.5−10%, *w*/*v*), temperature (110−130 °C), and time (5−30 min) using a one-factor-at-a-time (OFAT) approach. The highest TPC of 55.59 ± 0.12 µmole GAE/g _CS_ was achieved at 5.0% (*w*/*v*) CS loading and autoclaving at 125 °C for 25 min. Following hydrothermal extraction, CS protein was extracted from HT-extracted solid fraction by microwave-assisted alkaline extraction (MAE) using 0.2 M NaOH at 90 W for 2 min, resulting in a protein recovery of 12.19 ± 0.39 mg/g _CS_. The CS protein was then subjected to enzymatic hydrolysis using protease from *Bacillus halodurans* SE5 (protease_SE5). Protease_SE5-derived CS protein hydrolysate had a peptide concentration of 0.73 ± 0.09 mg/mL, with ABTS, DPPH, and FRAP values of 15.71 ± 0.10, 16.63 ± 0.061, and 6.48 ± 0.01 µmole TE/mL, respectively. Peptide identification by LC-MS/MS revealed several promising biological activities without toxicity or allergenicity concerns. This study’s integrated approach offers a sustainable and efficient method for extracting valuable compounds from CS, with potential applications in the food and pharmaceutical industries.

## 1. Introduction

Coffee is globally renowned as one of the most favored beverages [1]. To produce roasted coffee beans, green coffee beans are subjected to the roasting process [2], which generates 0.75% (*w*/*w*) of coffee silverskin (CS) [1]. Currently, the majority of CS is discarded in landfills, posing a significant environmental impact [3]. CS is composed of cellulose (18%), hemicellulose (13%), protein (19%), and minerals (8%) [4], as well as phenolic compounds. Among them, phenolic compounds have shown biological potential and can serve as a starting material for developing a new source of bioactive compounds.

Traditionally, phenolic compounds are extracted from CS and other plant materials using organic solvents such as methanol, ethanol, and their acidified derivatives [5,6], or aqueous acetone. However, there are many concerns, such as health, safety, and flammability, which limit their use in food applications [7]. In contrast, hydrothermal or pressurized hot water extraction offers a greener alternative method for extraction of natural compounds. For instance, Conde et al. (2016) [8] conducted hydrothermal extraction of CS using mild pretreatment at 120 °C for 20 min, yielding 19.17 mg of gallic acid equivalents/g _CS_. Similarly, Procentese et al. (2018) [9] suggested that hydrothermal extraction at 125 °C for 60 min resulted in a total phenolic compound content of 12.1 mg of gallic acid equivalents/g _CS_.

Another unexplored component of CS is protein. Even though alkaline extraction methods give high protein yields, strong alkaline conditions might affect protein bioactivity and their functional properties. To overcome these drawbacks, novel extraction methods, such as ultrasound-assisted alkaline extraction (UAE) and microwave-assisted alkaline extraction (MAE), have been proposed. UAE enhances extraction efficiency by leveraging cavitation phenomena to disrupt cells and membranes, while MAE increases temperature and pressure to facilitate compound extraction. Both UAE and MAE can reduce the alkaline concentration, which reduces the possibility of damaging proteins. To enhance the biological activities of proteins, the extracted proteins need to be hydrolyzed by enzymatic hydrolysis using various types of proteases. For instance, enzymatic hydrolysis of keratin waste by a keratinase from *Thermoactinomyces vulgaris* TK1-21 resulted in keratin hydrolysate with increased DPPH and ABTS values [10]. In addition, bioactivity of sericin extract [11] and CS protein [12] were increased after proteolytic hydrolysis by alkaline protease from *B. halodurans* SE5 and Alcalase, respectively.

This study focuses on the development of a novel extraction method for phenolic compounds and proteins with biological properties. The integrated process involves hydrothermal extraction of phenolic compounds, followed by extraction of protein from CS, and the production of bioactive peptides from CS protein using a protease from *B. halodurans* SE5 (protease_SE5). This work also aims to explore a sustainable waste management solution to transform CS residue into valuable compounds, in line with eco-friendly practices, the circular economy, and zero waste management.

## 2. Materials and Methods

### 2.1. Optimization of Hydrothermal Extraction of Phenolic Compounds from CS

Dried CS, provided by the Department of Animal and Aquatic Sciences, Faculty of Agriculture, Chiang Mai University, Thailand, was milled and stored in sealed plastic bags at 4 °C in darkness until further use. The particle size and moisture content of CS were 100 mesh and 5.60 ± 0.23% (*w*/*w*), respectively. The hydrothermal extraction of phenolic compounds from CS was optimized using a one-factor-at-a-time (OFAT) approach. Three independent parameters, including CS loading, temperature, and extraction time, were varied in the ranges of 2.5−15.0% (*w*/*v*), 110−135 °C, and 5−30 min, respectively. The extraction process was carried out using deionized water. All experiments were conducted in an autoclave (Purister-80, Cryste, Korea). The mixture was filtered using Whatman no.1 filter paper, resulting in two fractions, the HT-extracted CS liquid and solid fractions. The HT-extracted CS was then analyzed for antioxidant activity and total phenolic content (TPC). Meanwhile, the solid fraction was subjected to a protein extraction process.

### 2.2. Identification of Phenolic Compounds and Derivatives by LC-QTOF-MS

The HT-extracted CS was prepared using the optimal extraction conditions obtained from the OFAT approach. The liquid samples were dried using a freeze dryer and subjected to analysis by LC-quadrupole-time-of-flight mass spectrometer (LC-QTOF-MS). Identification of phenolic compounds and derivatives in HT-extracted CS was performed at the Science and Technology Research Institute, Chiang Mai University, Thailand [13]. The samples were analyzed using an Agilent 1290 Infinity II series coupled with a 6546 LC-QTOF instrument. Electrospray ionization (ESI) was used for ionization in negative mode. The nebulizer was operated at 35 psi with 11 L/min N_2_ flow. The capillary temperature was kept at 350 °C, while the sample flow rate was set at 0.2 mL/min. The m/z range was scanned within 100−1700, the capillary voltage was 3500 V, and the dry heater temperature was 320 °C. Provisional compound identification was performed by matching accurate mass results to content from the METLIN Personal Compound Database and Library (PCDL).

### 2.3. Effect of Extraction Methods on Extraction of Protein from CS

#### 2.3.1. Conventional Alkaline Extraction (CAE)

Conventional alkaline extraction (CAE) was investigated according to the modified methods [14]. The HT-extracted CS solid fraction was mixed with 0.2−1.0 M NaOH at 10.0% (*w*/*v*). Protein was extracted at 50 °C for 240 min and at 90 °C for 30 min. The supernatant was separated by centrifugation at 6000 rpm (4430× *g*) for 20 min. Subsequently, protein was harvested by precipitation. The sample was adjusted to pH 3.5 using 6.0 M HCl, and the precipitant was separated by centrifugation at 6000 rpm (4430× *g*) for 20 min.

#### 2.3.2. Ultrasound- and Microwave-Assisted Alkaline Extraction 

Ultrasound-assisted alkaline extraction (UAE) was conducted using an ultrasonic generator (V505, Sonics & Materials BE, Newtown, CT, USA) connected with a 1.8 cm diameter probe (20 kHz). The HT-extracted CS solid fraction was mixed with 0.2 M NaOH solution at 10.0% (*w*/*v*). Ultrasound treatment was performed at an amplitude of 80%, with on/off time intervals of 2 and 5 s, for 10−40 min. The mixture was centrifuged at 6000 rpm (4430× *g*) to obtain supernatant for protein precipitation. For microwave-assisted alkaline extraction (MAE), the HT-extracted CS solid fraction was immersed in 0.2 M NaOH at 10.0% (*w*/*v*). Afterwards, the mixture was subjected to an electric power of 90 W for 2−10 min. The supernatant was separated by centrifugation at 6000 rpm (4430× *g*) for 20 min. Subsequently, the supernatants from both UAE and MAE were adjusted to pH 3.5 using 6.0 M HCl for protein precipitation.

### 2.4. Enzymatic Production of Bioactive Peptides

To enhance biological activity, the proteins extracted from CS by CAE, UAE, and MAE were used as the substrate for bioactive peptide production using protease_SE5. The enzyme was produced according to a previous study [15]. The production medium contained 1.0 g/L K_2_HPO_4_, 0.5 g/L NaCl, 0.1 g/L CaCl_2_·2H_2_O, 0.1 g/L MgSO_4_·7H_2_O, and 0.2 g/L Tween 80, supplemented with 10 g/L urea as an enzyme inducer. After autoclaving, the medium was adjusted to pH 9.70. The enzyme production was conducted in a 5 L stirred tank bioreactor (BE Marubishi, Pathum Thani, Thailand) at 45 °C, an aeration rate of 2.0 vvm, and agitation rate of 240 rpm. Prior to use, crude protease_SE5 was dialyzed against 0.01 M Tris-HCl buffer, pH 7.5, at 4 °C for 12 h. The precipitated CS protein solution was prepared in 0.05 M sodium carbonate bicarbonate buffer, pH 9.5, at a concentration of 5 mg/mL. To start enzymatic hydrolysis, dialyzed protease_SE5 (200,000 U/g protein) was added, and the mixture was incubated at 55 °C for 12 h. Subsequently, the supernatant was collected by centrifugation at 10,000 rpm (9500× *g*). The CS protein hydrolysate was analyzed for molecular weight distribution, protein and peptide concentrations, and antioxidant activity. Additionally, the efficiency of protease_SE5 was compared with that of commercial alkaline protease (Alcalase) (Sigma-Aldrich, St. Louis, MO, USA).

### 2.5. Fractionation and Identification of Bioactive Peptides

The molecular weight (MW) distribution of low MW proteins and peptides was determined by tricine-SDS-PAGE using a 16.5% separating gel, and the bands were visualized by Coomassie brilliant blue G-250 [16]. In addition, samples were fractionated using a modified method based on that described by Yakul et al. (2021) [15]. Initially, the hydrolysate solution was separated through a 3 kDa MW cut-off (MWCO) polyethersulfone (PES) membrane (Merck, Kenilworth, NJ, USA). Subsequently, the <3 kDa fraction was lyophilized and dissolved in deionized water at a concentration of 25 mg/mL. This solution was then filtered using a 0.22 μm cellulose acetate membrane (Labfil, Shaoxing, Zhejiang, China), and 1 mL of the sample was applied to a Sephadex G-25 (GE Healthcare, Uppsala, Sweden) column (1.5 × 75 cm, column volume 120 mL). Fractions (2.5 mL/tube) were collected through elution with deionized water at a flow rate of 23 mL/h, and pooling was performed based on the absorbance at 280 nm. The pooled fractions were subsequently lyophilized. The bioactivity of each lyophilized fraction was evaluated, and those fractions exhibiting antioxidant activity were subjected to LC-MS/MS analysis at the Salaya Central Instrument Facility, Mahidol University, Thailand. The MS/MS data were subjected to analysis by PEAKS Xpro software (Bioinformatics Solutions Inc., Waterloo, ON, Canada). Protein and peptide concentrations prepared from each process were determined using Bradford [17] and ninhydrin [18] methods using BSA and glycine as the standards. Furthermore, the antioxidant capacity of the active fractions was assessed.

### 2.6. Analytical Methods

#### 2.6.1. Assays for Antioxidant Activity

The 2,2′-diphenyl-1-picryl-hydrazyl (DPPH) radical scavenging activity was determined following the method described by Veenashri and Muralikrishna [19], with some modifications. Samples were mixed with an ethanolic DPPH solution (80 mg/L) for 60 min before measuring the absorbance at 517 nm. For the 2,2′-azino-bis (3-ethylbenzothiazoline-6-sulfonic acid) (ABTS) radical scavenging activity, the modified method of Shazly et al. (2017) [20] was employed. A working solution of ABTS radical (ABTS^+^) was prepared by mixing 2.45 mM potassium persulfate solution with 7 mM ABTS solution (1:1, *v*/*v*) and incubating in the dark for 12–16 h. The reaction mixture containing 0.1 mL of the sample and 1.7 mL of the prepared ABTS^+^ solution was incubated at 37 °C in the dark for 30 min, and the radical scavenging activity was determined by measuring the decrease in A_734_. Additionally, the ferric-reducing antioxidant power (FRAP) assay was conducted as described by Benzie and Strain (1996) [21]. A stock solution was prepared containing 300 mM acetate buffer (pH 3.6), 20 mM FeCl_3_·6H_2_O, and 10 mM TPTZ in 40 mM HCl (10:1:1, *v*/*v*/*v*). Then, 0.1 mL of the sample was added to 0.9 mL of FRAP reagent, and the mixture was incubated at 37 °C in the dark for 30 min. Absorbance was measured at 595 nm. Antioxidant activity was calculated as μmol Trolox equivalents (TE)/mL.

#### 2.6.2. Determination of Total Phenolic Content (TPC)

The total phenolic content composition in each sample were determined by the Folin–Ciocalteu colorimetric method [22]. In brief, 0.3 mL of sample was mixed with 1.5 mL of 0.2 M Folin–Ciocalteu phenol reagent. The reaction was allowed to incubate for 5 min. Then, 1.2 mL of 0.7 M sodium carbonate (Na_2_CO_3_) solution was added to the reaction, and the solution was mixed well. The reaction tubes were incubated at 37 °C in the dark for 2 h. The absorbance at 760 nm was measured. The calibration curve was prepared by using gallic acid as a standard with the working range of 0–600 µmole/mL (y = 0.0018x, R^2^ = 0.9994). The results are reported as µmole GAE/mL.

### 2.7. Statistical Analysis

Experiments were carried out in triplicate. Statistical analysis was performed by one-way analysis of variance (ANOVA; *p* < 0.05) using SPSS Statistics for Windows v.17.0 (SPSS Inc., Chicago, IL, USA) with Duncan’s multiple range test.

## 3. Results

### 3.1. Optimization of Hydrothermal Extraction of Phenolic Compounds from CS

To optimize the extraction conditions, this study investigated the effects of CS loading (2.5−15.0% *w*/*w*), temperature (110−135 °C), and extraction time (5−30 min) on extraction of phenolic compounds using an OFAT approach. Figure 1a demonstrates that CS loading significantly influenced TPC and the antioxidant activity of phenolic compounds from CS. The optimal CS loading was 5% (*w*/*v*). TPC reached 24.41 ± 0.16 µmole GAE/g _CS_, with ABTS, DPPH, and FRAP values of 58.71 ± 33.04, 33.04 ± 0.08, and 35.98 ± 0.46 µmole TE/g _CS_, respectively. Temperature also affected the extraction of phenolic compounds from CS, and the optimal temperature was observed at 125 °C, with TPC, ABTS, DPPH, and FRAP values of 50.38 ± 0.31 µmole GAE/g CS, 65.34 ± 0.41, 35.20 ± 0.34, and 37.83 µmole TE/g _CS_, respectively (Figure 1b). Regarding extraction time, the highest total phenolic compound was obtained after extraction for 25 min. TPC reached 55.59 ± 0.12 µmole GAE/g _CS_, and the ABTS, DPPH, and FRAP values were 82.94 ± 1.95, 43.37 ± 0.24, and 43.26 ± 0.37 µmole TE/g _CS_, respectively (Figure 1c).

### 3.2. Identification of Phenolic Compounds and Derivatives by LC-QTOF-MS

The profile of phenolic compounds in the HT-extracted CS liquid fraction was analyzed by LC-QTOF-MS using untargeted screening and identification. The HT-extracted CS liquid fraction was prepared using CS loading, temperature, and extraction time of 5% (*w*/*v*), 125 °C, and 25 min, respectively. The identified compounds with matching scores of >70% were considered. Among these, alkaloids and various polyphenols, such as phenolic acids, flavonoids, xanthones, and secoiridoids, were observed (Table 1).

### 3.3. Enzymatic Production of Bioactive Peptides from CS

After extraction of phenolic compounds, the HT-extracted CS solid fraction was obtained and subjected to protein extraction by CAE using different concentrations of NaOH (0.2, 0.4, 0.6, 0.8, and 1.0 M). Two different conditions (50 °C for 240 min and 90 °C for 30 min) were compared in terms of protein recovery yield, peptide concentration, and antioxidant activity. As shown in Figure 2a,b, a correlation between the concentration of NaOH and the protein recovery yield was observed. However, subsequent hydrolysis with protease_SE5 revealed that the increase in peptide concentration was limited due to the use of a high concentration of NaOH, a strong alkaline condition. In addition, low peptide concentrations of 0.04 ± 0.01 and 0.03 ± 0.01 mg/mL were observed for strong alkaline conditions of 1.0 M NaOH at 50 °C for 240 min and 90 °C for 30 min, respectively. By contrast, using low alkaline conditions (0.2 M NaOH), the increase in peptide concentration was significantly higher, with values of 0.26 ± 0.01 and 0.20 ± 0.01 mg/mL at 50 °C for 240 min and 90 °C for 30 min, respectively. Additionally, utilizing low alkaline conditions (0.2 M NaOH) and mild temperature (50 °C for 240 min) was favorable in producing bioactive peptides from CS (Figure 2c), resulting in the highest antioxidant activities for ABTS (0.83 ± 0.01 µmole TE/mL), DPPH (0.89 ± 0.01 µmole TE/mL), and FRAP (0.61 ± 0.01 µmole TE/mL).

To improve extraction efficiency and reduce extraction time, physical extraction methods, including ultrasound and microwave, were combined with CAE under low alkaline conditions (0.2 M NaOH). UAE was performed at 20 kHz, at an amplitude 80%, for extraction times of 10, 25, and 40 min (Table 2). The protein concentrations obtained were 13.06 ± 0.71, 18.31 ± 0.22, and 18.60 ± 0.36 mg/g _CS_, respectively. Following hydrolysis with protease_SE5, the antioxidant activities increased and were within the ranges of ABTS (0.50–1.07 µmole TE/mL), DPPH (0.26–1.03 µmole TE/mL) and FRAP (0.04–0.60 µmole TE/mL).

Furthermore, MAE was performed using a microwave power of 90 W and extraction durations of 2, 5, and 10 min with 0.2 M NaOH. The obtained results indicated protein recovery yields of 12.19 ± 0.39, 20.06 ± 0.39, and 21.77 ± 0.32 mg/mL, respectively. Although the protein recovery yield from MAE for 2 min was comparable to that from CAE (12.26 ± 0.96 mg/mL), subsequent hydrolysis with protease_SE5 revealed that the increase in peptide concentration was relatively high. Additionally, the antioxidant activity of the CS protein hydrolysate from MAE was significantly higher than that from CAE and UAE. It should be noted that MAE at a microwave power of 90 W for 2 min of extraction time not only improved the extraction efficiency but also reduced the process time from 240 to 2 min. 

Furthermore, the molecular weight (MW) distribution of bioactive peptides obtained from different extraction processes was investigated using tricine-SDS-PAGE. Before enzymatic hydrolysis of CS protein, MW was in the range of 10 to 37 kDa (Figure 3). At 12 h, protease_SE5 effectively hydrolyzed high MW peptides, resulting in peptides with a lower MW (≤5 kDa). The SDS-PAGE results were relevant to the antioxidant values (Table 3). Notably, the highest intensity band of <3 kDa was observed for MAE.

### 3.4. Comparative Study of Bioactive Peptide Production from CS by Protease_SE5 and Alcalase

The efficiency of bioactive peptide production from CS by protease_SE5 was compared to that by commercial alkaline protease (Alcalase). The CS protein was extracted by MAE at 90 W for 2 min using 0.2 M NaOH and used as a substrate for bioactive peptide production (CS protein hydrolysate). After hydrolysis for 12 h, the bioactive peptides generated from protease_SE5 showed a high peptide concentration of 0.73 ± 0.09 mg/mL, as well as ABTS, DPPH, and FRAP values of 15.71 ± 0.10, 16.63 ± 0.06, and 6.48 ± 0.01 µmole TE/mL, respectively (Figure 4). By contrast, the bioactive peptides produced by Alcalase displayed a lower peptide concentration of 0.58 ± 0.01 mg/mL, along with ABTS, DPPH, and FRAP values of 13.27 ± 0.36, 13.82 ± 0.09, and 5.71 ± 0.21 µmole TE/mL, respectively. The CS protein hydrolysates from both enzymes were subsequently identified through LC-MS/MS analysis.

### 3.5. Fractionation and Identification of Bioactive Peptide

The CS protein hydrolysate was fractionated by ultrafiltration (UF) using a 3 kDa MWCO PES membrane. Subsequently, the <3 kDa fraction was further processed through a Sephadex G-25 column for desalting and fractionation of protein hydrolysates. The CS protein hydrolysates from both protease_SE5 and Alcalase were divided into five peaks (Figure 5a,b). However, fractions 3 (F3), 4 (F4), and 5 (F5) were subsequently subjected to LC-MS/MS due to their high antioxidant activity (Figure 5c,d).

The peptide sequences were identified by *de novo* sequencing. Peptides with ALC ≥ 90% and local confidence score ≥ 80% were considered. However, peptides identified from F5 were excluded because the ALC score was lower than 90%. Eight peptides were observed for the CS protein hydrolysate from protease_SE5. For the CS protein hydrolysate from Alcalase, six peptides were obtained. These identified peptides were comprised of 4−9 amino acid residues with MW of 448−1051 Da (Table 3). 

Table 4 shows the percentage of key antioxidative amino acids and analysis of the identified peptides by several web servers. According to previous studies, the proposed mechanism for antioxidant peptides involves scavenging free radicals by inhibiting electron migration, hydrogen atom transfer, or transient metal ion chelation. Six amino acids were proven to be key residues involved in antioxidant activity, including cysteine (Cys, C), histidine (His, H), methionine (Met, M), tryptophan (Trp, W), tyrosine (Tyr, Y), and phenylalanine (Phe, F). These amino acids function as direct radical scavengers by donating hydrogen to neutralize unpaired electrons or radicals [23,24]. In this study, the identified peptides had percentages of key constituent amino acids in the range of 32.70−66.80%. According to the analysis of identified peptides by BIOPEP, various biological activities, such as angiotensin-converting enzyme (ACE) inhibition, dipeptidyl peptidase IV (DPP-4) inhibition, and antioxidant properties, were predicted. In addition, all peptides showed a Peptide Ranker (http://distilldeep.ucd.ie/PeptideRanker (accessed on 23 February 2024)) score of >0.5, suggesting high probability of being bioactive peptides. Based on the prediction by ToxinPred, the bioactive peptides from CS were non-toxic.

## 4. Discussion

CS is derived from the coffee roasting process. The growing demand for coffee consumption will generate CS, which may cause a serious impact on the environment. Previous studies have reported the way to deal with several wastes from different sectors, such as blueberry by-products, pine nut skin, sea bass by-products, and plant by-products from banana harvesting [25,26,27,28]. After valorization, valuable compounds are obtained and can be applied in food applications. CS consists of various interesting bioactive compounds, and it could be a good source of protein and polyphenols [1,29,30,31]. Moreover, a recent report noted that CS did not show carcinogenic risk. Therefore, CS is safe to be valorized and used as an alternative ingredient for functional foods and pharmaceutical applications [29]. Here, the integrated process of phenolic compound extraction and bioactive peptide production from CS was demonstrated. The first step involves extraction of phenolic compounds from CS by HT, which requires water as an extraction solvent in combination with high temperature and pressure [31]. Key factors affecting the extraction efficiency of HT are solid loading, temperature, and extraction time. The results showed that CS loading affected the efficiency of extraction of phenolic compounds from CS. The use of CS loading above 5.0% (*w*/*v*) lowered the antioxidant activity and TPC (Figure 1). This could be explained by mass transfer ability and penetration of water into the tested sample. A sufficient solvent ratio could promote the desorption of target compounds from the CS into the extraction solvent [14]. Temperature is also one of the crucial factors affecting the extraction efficiency. Obviously, the optimal temperature was 125 °C. Further increasing the temperature led to decreases in TPC and antioxidant activity. Temperatures above 125 °C can cause Maillard reactions, resulting in the degradation of phenolic compounds and affecting the antioxidant activity of the HT-extracted CS liquid fraction. The results were in accordance with a previous study in which an extraction temperature <200 °C did not negatively affect the phenolic compounds and antioxidant activity of CS extracts. However, TPC and antioxidant activity decreased when the extraction temperature was above 200 °C [30]. To deeply investigate the bioactive compounds in the HT-extracted CS liquid fraction, the sample was extracted under optimized conditions and subjected to LC-QTOF-MS analysis. The results revealed that the HT-extracted CS liquid fraction contained diverse bioactive compounds, including alkaloids, polyphenols, and flavonoids (Table 1). Previous studies also reported that caffeine and chlorogenic acid (CGA) were found in CS extract. CGA is the ester of caffeic acid and quinic acid. In addition, caffeoylquinic acid (CQA) is the isomer of CGA in CS [32]. CQA is the most abundant alkaloid and phenolic acid found in CS extract [30]. These phenolic compounds are well known as bioactive compounds due to their biological activities such as antioxidant activity, radioprotection, and anti-neuroinflammation [33,34]

In addition, CS can be used as an alternative source of protein [35,36]. Thus, the second step of an integrated process is the extraction of protein from CS for bioactive peptide production. Enzymatic hydrolysis by proteases is an interesting process to generate bioactive peptides due to the high specificity of the enzymes and mild conditions required [1]. In general, the substrate for enzymatic production of bioactive peptides should be high quality or non-denatured protein. The extraction method is a primary concern to obtain high-quality protein. In this work, the efficiency of different extractions, including CAE, MAE, and UAE, was compared in terms of both protein recovery and the quality of extracted CS protein for bioactive peptide production. CAE is an extraction method that is commonly used for protein extraction at an industrial scale because of high protein yield and recovery [37]. However, the major drawback of CAE is the requirement of a high concentration of alkaline solution, which could have a negative effect on protein structure and functionality. As clearly evidenced in Figure 2, high protein recovery yield was achieved using a higher concentration of NaOH or strong alkaline conditions. However, the CS protein was not suitable for protein hydrolysis due to the low concentration of peptides and antioxidant activity of the CS protein hydrolysate. Previous studies also indicated that a high concentration of NaOH could improve protein recovery, but it also caused denaturation of the protein structure [38,39,40], making the extracted protein substrate unsuitable for production of bioactive peptides. In comparison to the strong alkaline conditions, extraction under low alkaline conditions using 0.2 M NaOH followed by enzymatic hydrolysis by protease_SE5 could produce CS protein hydrolysate with a higher peptide concentration and antioxidant activity. As a result, the alkaline concentration of 0.2 M was chosen for further studies.

Furthermore, extracting protein from CS by CAE at 50 °C was more efficient than extraction at a higher temperature (90 °C), as indicated by a higher peptide concentration and increased antioxidant activity of the CS protein hydrolysate. Nevertheless, the extraction time was still too long (240 min). To improve the efficiency of CAE, ultrasound (UAE) and microwave (MAE) were introduced to CAE to assist in the extraction of CS protein (Table 2). The influence of UAE on the extraction process can be elucidated by cavitation phenomena. The collapse of bubbles enhances the penetration of alkaline solution into the internal structure and consequently promotes mass transfer of CS protein, enhancing the yield [39]. Meanwhile, MAE involves the use of electromagnetic energy at a specific power, which is transferred to the location of the compounds of interest in the substrate, leading to rapid heat in short processing times. As mentioned above, high temperature affects the protein structure and quality [36]. Therefore, MAE was applied for 2–10 min at a low microwave level of 90 W to avoid heat accumulation. Interestingly, the yield of CS protein extracted by MAE at 90 W for 10 min (21.76 mg/g _CS_) was 1.7-fold higher than that by UAE (13.06 mg/g _CS_) using the same extraction time. Microwave irradiation can enhance the extraction efficiency by greatly shortening the extraction time [39,40]. Moreover, the antioxidant activity of the CS protein hydrolysate derived from MAE for 2 min was higher than that of the CS protein hydrolysate from CAE as well as from UAE. The peptide concentration was found to be 1.3-fold higher than that in the CS protein hydrolysate from CAE and UAE. 

To investigate whether the MW of the proteins changed, the MW distribution of CS protein hydrolysate was performed by tricine-SDS-PAGE. After 12 h of hydrolysis by protease_SE5, the change from high MW proteins (15−37 kDa) to lower MW peptides (≤ 10 kDa) could be observed clearly. Proteases break down larger protein molecules into smaller peptide fragments by the specific cleavage of peptide bonds within the protein substrate [41]. At 0 h, the high MW protein band had a high intensity, indicating the presence of intact proteins in the CS protein before enzymatic hydrolysis. After hydrolysis for 12 h, the intensity of high MW bands decreased, while new bands corresponding to lower MW peptides appeared. The changes in band pattern confirmed successfully proteolytic hydrolysis of CS protein by protease_SE5. It is well known that peptides with a lower MW exhibit better susceptibility to interaction with free radicals in *in vitro* assays [42]. The results indicated that MAE not only required low electric power and relatively short extraction time but also provided high-quality CS protein and CS protein hydrolysate with a higher antioxidant activity compared to UAE and CAE. 

Protease_SE5, an in-house enzyme derived from *B. halodurans* SE5, has demonstrated remarkable efficiency in hydrolyzing proteins to enhance antioxidant activity [15]. Meanwhile, commercial alkaline protease (Alcalase) is widely employed for protein hydrolysis [23,24,43]. Herein, a comparative study of bioactive peptide production from CS protein by protease_SE5 and Alcalase was conducted under the same conditions. As depicted in Figure 4, the antioxidant activity of CS protein hydrolysate improved after enzymatic treatment by both proteases. Notably, CS protein hydrolysate from protease_SE5 exhibited higher antioxidant activity values than CS protein hydrolysate produced from Alcalase. This difference in antioxidant value might be attributed to several factors. Different proteases exhibit different degrees of substrate specificity based on their amino acid sequence preferences [44]. Protease_SE5 probably had a better compatibility with CS protein, leading to more efficient hydrolysis. 

To further investigate if there were differences in the peptide sizes and sequences of the CS protein hydrolysates derived from two different enzymes, CS protein hydrolysate were fractionated by UF membrane and Sephadex G-25 column. The active fractions were subjected to LC-MS/MS analysis for peptide identification. Based on separation by Sephadex G-25 (Figure 5), the chromatographic profiles of the two enzymes were similar in terms of A_280_ and ABTS radical scavenging activity. The three most active fractions, including F3, F4, and F5, were analyzed by LC-MS/MS. The obtained data were analyzed by PEAKS using *de novo* peptide sequencing. The identified peptides were filtered out by ALC score (Table 3). Generally, the ALC score is a measure used in mass spectrometry-based proteomics to assess the reliability of peptide identification. A higher ALC score typically indicates a more confident peptide identification, suggesting that the identified peptide is more likely to be correct [45]. It was found that the identified peptides were comprised of 4−9 residues. Normally, bioactive peptides contain 2–20 amino acids. The biological activity of a protein hydrolysate depends on the amino acid composition and their position because of the protein source and the specificity of the enzyme used [15]. Other than that, the identified peptides had high percentages of key amino acids, such as His, Phe, Trp, and Tyr (Table 4). Studies have highlighted the significant contribution of His, Phe, Trp, and Tyr to the antioxidant properties of peptides. These amino acids can act as radical scavengers by serving as hydrogen donors to neutralize unpaired electrons or radicals [23,24,44]. Furthermore, the biological potential was assessed using Peptide Ranker, BIOPEP, and ToxinPred tools. Peptides identified from both protease_SE5 and Alcalase exhibited diverse biological activities, including angiotensin-converting enzyme (ACE) inhibition, dipeptidyl peptidase IV (DPP-4) inhibition, and antioxidant capacity, with high scores of probability. Moreover, Nolasco et al. (2022) reported that the CS protein hydrolysate provides ACE inhibitory peptides, antioxidants, and DPP IV inhibitors, with possible uses as nutraceuticals or conventional drugs [1]. It should be noted that the bioactive peptides from CS could be an interesting source of ACE inhibitory antioxidants and dipeptidyl peptidase (DPP) IV inhibitors. 

This study suggests that in-house protease_SE5 exhibits efficiency comparable to Alcalase, a commercial enzyme, in producing bioactive peptides from CS. Nevertheless, further *in vivo* studies are required to validate the bioactivity of the peptides.

## 5. Conclusions

The present study investigated the possibility of CS valorization by an integrated process. Hydrothermal extraction was successful in extracting phenolic compounds from CS. Under the optimal conditions, TPC of 55.59 ± 0.12 µmole GAE/g _CS_ with high antioxidant activity was achieved at 5.0% (*w*/*v*) CS loading with autoclaving at 125 °C for 25 min. Moreover, the CS protein was extracted by MAE at 90 W for 2 min using 0.2 M NaOH, resulting in high protein quality. Bioactive peptides from CS protein extracted by MAE had higher antioxidant activity in comparison to those extracted by UAE and CAE. In addition, identification of peptides by LC-MS/MS demonstrated the biological potential as ACE inhibition and DPP-4 inhibition. Therefore, the present work shows that an integrated process might be used for sustainable valorization of CS, transforming it into valuable compounds for functional food and pharmaceutical applications.

## Figures and Tables

**Figure 1 foods-13-01230-f001:**
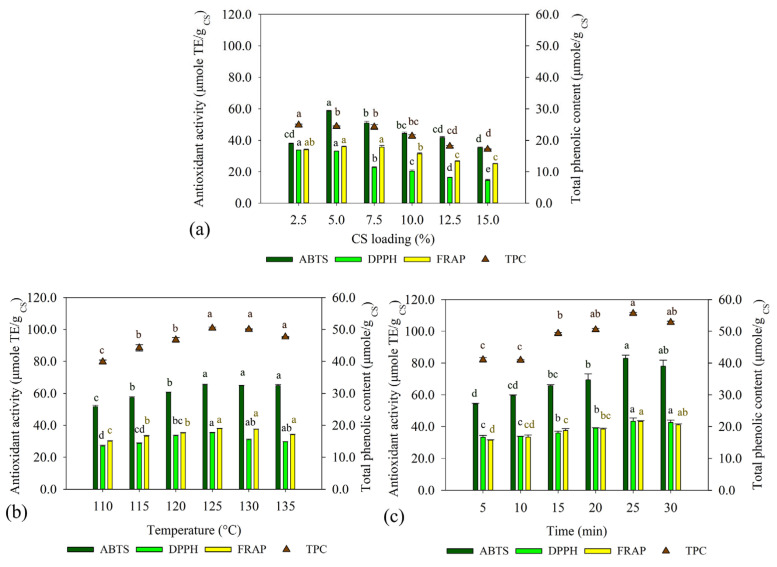
The antioxidant activity and TPC of the HT-extracted CS liquid fraction. The effects of (**a**) CS loading (% *w*/*v*), (**b**) temperature (°C), and (**c**) time (min) on antioxidant activity and TPC were investigated. The experiments are performed in triplicate (*n* = 3). The results are reported as mean  ±  SD. Different letters (a–e) indicate significant differences at *p*  <  0.05 according to analysis by Duncan’s multiple range test.

**Figure 2 foods-13-01230-f002:**
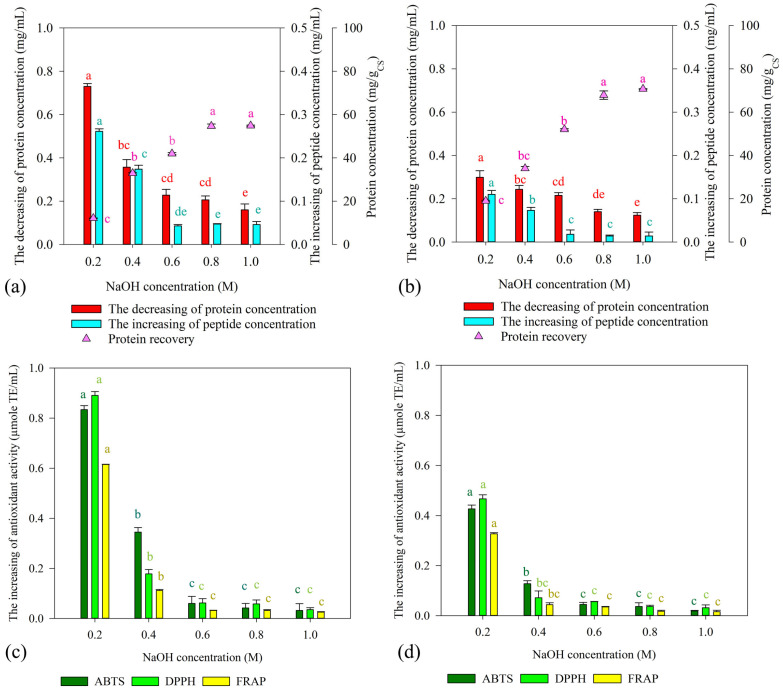
Enzymatic production of bioactive peptides from CS. Protein recovery, decrease in protein concentration, and increase in peptide concentration by CAE at 50 °C for 240 min (**a**) and 90 °C for 30 min (**b**). The increase in antioxidant activity of CS protein hydrolysate derived from CAE at 50 °C for 240 min (**c**) and 90 °C for 30 min (**d**). The protein from CS was extracted by CAE, followed by enzymatic hydrolysis using protease_SE5 (200,000 U/g protein) at pH 9.5 and 55 °C for 12 h. Different letters (a–e) indicate significant differences at *p*  <  0.05 according to analysis by Duncan’s multiple range test.

**Figure 3 foods-13-01230-f003:**
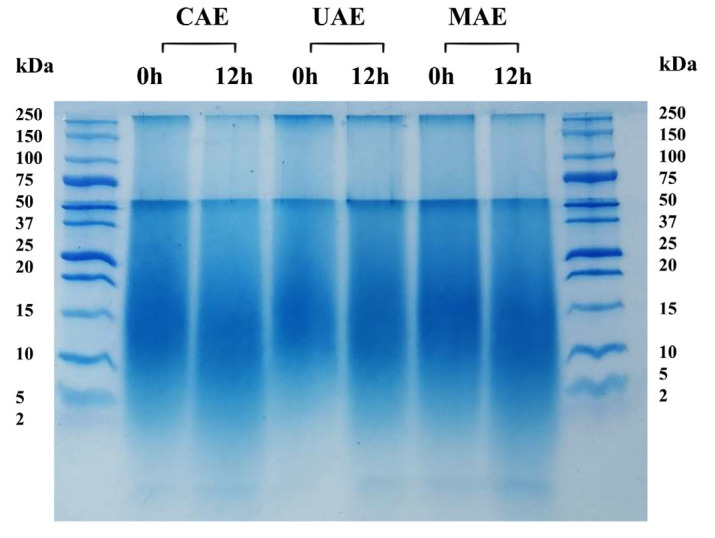
The molecular weight distribution of CS protein hydrolysate by tris-tricine SDS-PAGE analysis. Electrophoresis was carried out under denaturing conditions in 16.5% polyacrylamide gel. The CS protein was extracted by 0.2 M NaOH at 50 °C for 240 min (CAE), ultrasound-assisted alkaline extraction (UAE) for 10 min, and microwave-assisted alkaline extraction (MAE) at 90 W for 2 min. The enzymatic hydrolysis was performed at pH 9.5 and 55 °C for 12 h using protease_SE5 (200,000 U/g protein).

**Figure 4 foods-13-01230-f004:**
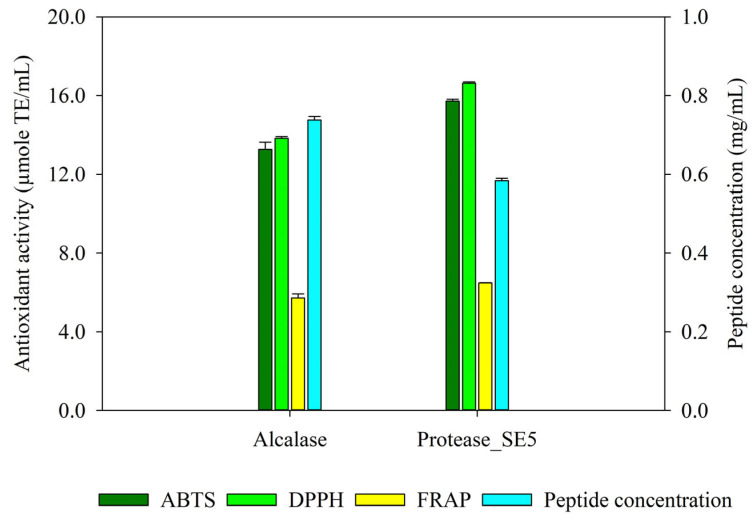
Peptide concentration and antioxidant activity of CS protein hydrolysate. The antioxidant activity of CS protein hydrolysate by ABTS, DPPH, and FRAP assay. The CS protein was extracted by MAE at 90 W for 2 min using 0.2 M NaOH and subjected to hydrolysis by either protease_SE5 or Alcalase (200,000 U/g protein) at pH 9.5 and 55 °C for 12 h.

**Figure 5 foods-13-01230-f005:**
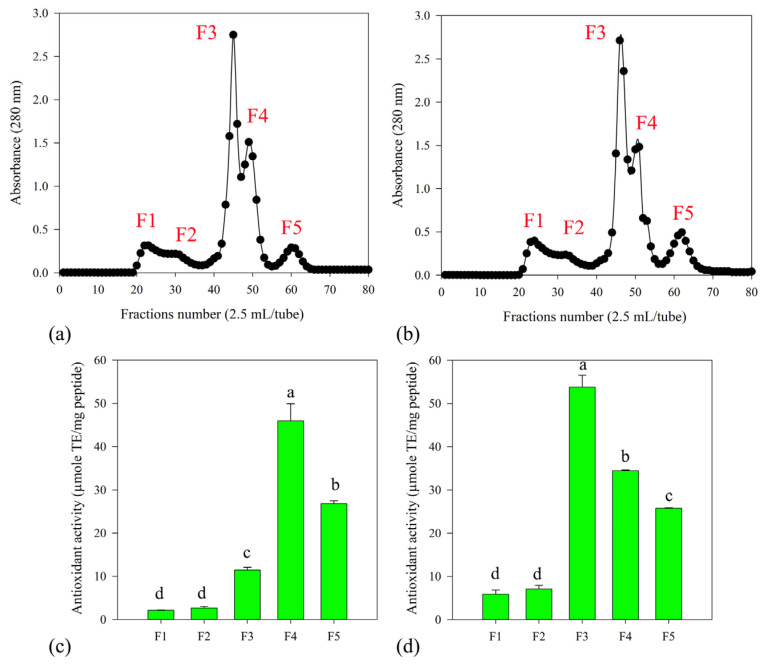
Size exclusion chromatography of <3 kDa from protease_SE5 (**a**) and Alcalase (**b**). Each fraction (2.5 mL) was eluted by deionized water at a flow rate of 23 mL/h. The active fractions were lyophilized before analysis by LC-MS/MS and *de novo* peptide sequencing. Antioxidant activity of active fractions from protease_SE5 (**c**) and Alcalase (**d**) obtained from the Sephadex G-25 column. The experiments are performed in triplicate (*n* = 3). Different letters (a–d) indicate significant differences at *p*  <  0.05 according to analysis by Duncan’s multiple range test.

**Table 1 foods-13-01230-t001:** List of phenolic compounds in HT-extracted CS liquid fraction identified by LC-QTOF-MS.

No.	Proposed Compounds	Molecular Formula	Mass	Matching Score (%) *
	**Alkaloids**			
1	Caffeine	C_8_H_10_N_4_O_2_	194.0795	74.20
	**Polyphenols**			
	**Phenolic acids**			
2	3-O-Caffeoylshikimic acid	C_16_H_16_O_8_	336.0838	95.70
3	5-O-Caffeoylshikimic acid	C_16_H_16_O_8_	336.0840	95.36
4	3-Dehydroshikimic acid	C_7_H_8_O_5_	172.0362	86.71
5	5Z-Caffeoylquinic acid CQA	C_16_H_18_O_9_	354.0943	94.83
6	Caffeic acid	C_9_H_8_O_4_	180.0419	93.69
7	*cis-p*-Coumaric acid-4-4-(apiosyl-(1->2)-glucoside)	C_20_H_26_O_12_	458.1412	81.76
8	Ferulic acid	C_10_H_10_O_4_	194.0579	97.13
9	Loganic acid	C_16_H_24_O_10_	376.1366	72.74
10	*p*-Coumaric acid	C_9_H_8_O_3_	164.0471	86.48
11	Quinic acid	C_7_H_12_O_6_	192.0629	97.79
12	Shikimic acid	C_7_H_10_O_5_	174.0526	94.58
13	Syring acid	C_9_H_10_O_5_	198.0531	97.57
14	*trans-p*-Coumaric acid-4-glucoside	C_15_H_18_O_8_	326.0992	89.13
15	Vanillic acid	C_8_H_8_O_4_	168.0421	90.81
	**Flavonoids**			
16	Kaempferol	C_15_H_10_O_6_	286.0464	75.89
17	Epicatechin	C_15_H_14_O_6_	290.0783	97.19
18	Quercetin	C_15_H_10_O_7_	302.0443	89.32
19	*trans*-Resveratrol	C_14_H_12_O_3_	228.0787	70.37
	**Xanthone**			
20	Isogentisin	C_14_H_10_O_5_	258.0542	87.97
	**Secoiridoids**			
21	Amarogentin	C_29_H_30_O_13_	586.1710	80.19

* Matching score (%) represents the correlation quality of precursor and corresponding fragment ion peaks for each compound.

**Table 2 foods-13-01230-t002:** Protein recovery, protein concentration, peptide concentration, and antioxidant activity of CS protein hydrolysate.

Extraction Time (min)	Protein Recovery (mg/g _CS_)	Decrease in Protein Concentration (mg/mL)	Increase in Peptide Concentration (mg/mL)	Increase in Antioxidant Activity (µmole TE/mL)		
				ABTS	DPPH	FRAP
Conventional alkaline extraction (CAE) *						
240	12.26 ± 0.96 ^d^	0.730 ± 0.017 ^b^	0.261 ± 0.006 ^a^	0.834 ± 0.016 ^a^	0.891 ± 0.015 ^a^	0.615 ± 0.001 ^a^
Ultrasound-assisted alkaline extraction (UAE) **						
10	13.06 ± 0.71 ^d^	0.879 ± 0.005 ^ab^	0.269 ± 0.009 ^a^	1.073 ± 0.015 ^a^	1.038 ± 0.008 ^a^	0.607 ± 0.009 ^a^
25	18.31 ± 0.22 ^c^	0.733 ± 0.011 ^b^	0.240 ± 0.004 ^b^	0.752 ± 0.081 ^b^	0.640 ± 0.012 ^b^	0.224 ± 0.007 ^b^
40	18.60 ± 0.36 ^c^	0.679 ± 0.010 ^c^	0.177 ± 0.052 ^c^	0.507 ± 0.030 ^c^	0.264 ± 0.009 ^c^	0.041 ± 0.006 ^c^
Microwave-assisted alkaline extraction (MAE) ***						
2	12.19 ± 0.39 ^d^	0.902 ± 0.090 ^a^	0.357 ± 0.020 ^a^	1.418 ± 0.002 ^a^	1.258 ± 0.051 ^a^	0.858 ± 0.052 ^a^
5	20.06 ± 0.39 ^b^	0.701 ± 0.016 ^b^	0.247 ± 0.008 ^b^	1.028 ± 0.028 ^b^	0.877 ± 0.051 ^b^	0.521 ± 0.002 ^b^
10	21.77 ± 0.32 ^a^	0.512 ± 0.042 ^d^	0.144 ± 0.001 ^d^	1.007 ± 0.035 ^b^	0.806 ± 0.006 ^b^	0.451 ± 0.021 ^b^

* Conventional alkaline extraction was performed at 50 °C. ** Ultrasound-assisted alkaline extraction was performed using 80% amplitude. *** Microwave-assisted alkaline extraction was performed using 90 W electric power. The experiments are performed in triplicate (*n* = 3). The results are reported as mean  ±  SD. Different letters (a–d) within columns are significantly different at *p*  <  0.05 according to analysis by Duncan’s multiple range test.

**Table 3 foods-13-01230-t003:** List of identified peptides from CS protein hydrolysate by LC-MS/MS coupled with *de novo* sequencing.

Peptide Sequence	ALC (%) *	Local Confidence (%)	m/z	Molecular Weight (Da)
**Protease_SE5**				
Fraction 3				
FLGY	90	93 84 88 93	499.2525	498.2478
FGGGF	94	92 86 96 97 98	484.2185	483.2118
FDYGKY	98	97 98 98 95 98 99	396.6815	791.3489
FYDTYY	93	80 89 96 96 99 99	436.1788	870.3436
YSYAYDDR	99	98 99 99 99 99 98 98 99	526.7186	1051.4246
Fraction 4				
YTRPY	98	99 99 99 98 97	350.1793	698.3387
YTEYAF	96	90 92 98 97 99 99	397.1747	792.3330
FDFVWVQ	96	92 95 79 83 99 99 99	397.1747	792.3330
**Alcalase**				
Fraction 3				
YTDHGAF	98	99 99 99 98 97 98 99	405.6722	809.3344
FGGGGSFPP	91	96 93 99 99 98 99 82 62 90	411.6886	821.3708
Fraction 4				
FGGY	91	95 86 90 93	443.1905	442.1852
FDYLR	93	94 96 93 90 93	357.1833	712.3544
DYFYY	92	88 81 95 97 99	385.6546	769.2959
WDAFHPT	93	89 93 95 98 97 92 88	437.1962	872.3817

* ALC (%) represents average local confidence.

**Table 4 foods-13-01230-t004:** Percentage of key antioxidative amino acids, prediction of biological activity, and toxicity of peptides identified from protease_SE5- and Alcalase-derived of CS protein hydrolysate.

Peptide Sequence	Key Constituent Antioxidative Amino Acids (%)						Peptide Ranker *	BIOPEP	
	Cys	His	Met	Phe	Trp	Tyr		Biological Activity **	Toxicity ***
**Protease_SE5**									
Fraction 3									
FLGY	0	0	0	16.7	0	16.7	0.91	ACE inhibitor, dipeptidyl peptidase IV inhibitor	Non-toxic
FGGGF	0	0	0	33.4	0	0	0.98	ACE inhibitor, dipeptidyl peptidase IV inhibitor	Non-toxic
FDYGKY	0	0	0	16.7	0	33.4	0.67	ACE inhibitor	Non-toxic
FYDTYY	0	0	0	16.7	0	33.4	0.60	ACE inhibitor, dipeptidyl peptidase IV inhibitor	Non-toxic
Fraction 4									
FDFVWVQ	0	0	0	33.4	16.7	0	0.68	ACE inhibitor, dipeptidyl peptidase IV inhibitor, antioxidant	Non-toxic
**Alcalase**									
Fraction 3									
YTDHGAF	0	16.7	0	16.7	0	16.7	0.40	dipeptidyl peptidase IV inhibitor	Non-toxic
FGGGGSFPP	0	0	0	33.4	0	0	0.84	ACE inhibitor, dipeptidyl peptidase IV inhibitor	Non-toxic
Fraction 4									
FGGY	0	0	0	16.7	0	16.7	0.95	ACE inhibitor	Non-toxic
FDYLR	0	0	0	16.7	0	16.7	0.81	ACE inhibitor	Non-toxic
DYFYY	0	0	0	16.7	0	50.1	0.84	ACE inhibitor, antioxidant	Non-toxic
WDAFHPT	0	16.7	0	16.7	16.7	0	0.84	ACE inhibitor, dipeptidyl peptidase IV inhibitor	Non-toxic

* Prediction of the probability that the peptide will be bioactive, available at http://distilldeep.ucd.ie/PeptideRanker/ (accessed on 23 February 2024). ** Prediction of the biological activity by BIOPEP, available at http://www.uwm.edu.pl/biochemia/index.php/pl/biopep (accessed on 23 February 2024). *** Prediction of toxicity of peptides by ToxinPred, available at https://webs.iiitd.edu.in/raghava/toxinpred/index.html (accessed on 23 February 2024).

## Data Availability

The original contributions presented in the study are included in the article, further inquiries can be directed to the corresponding author.

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
