# Peer review of "Sustainable Valorization of Coffee Silverskin: Extraction of Phenolic Compounds and Proteins for Enzymatic Production of Bioactive Peptides"

_foods, 2024, doi:10.3390/foods13081230_

Round 1

Reviewer 1 Report

Comments and Suggestions for Authors

The manuscript titled "Exploring the Sustainable Valorization of Coffee Silverskin: Extraction of Phenolic Compounds and Proteins for Enzymatic Production of Bioactive Peptide" discusses a study focused on developing innovative extraction methods for phenolic compounds and proteins from Coffee Silverskin (CS), a by-product of the coffee roasting process. Overall, the aims of the paper reflect a comprehensive approach to addressing environmental challenges through scientific innovation, with the potential to make significant contributions to the fields of waste management, sustainable agriculture, and green chemistry. 

Some observations:

1. The reference by Nolasco et al. is duplicated (1 and 33). Please check accordingly.

2. In some aspects, the article addresses the same points as Nolasco et al. [33], but this reference is scarcely used for an in-depth discussion of the results. This is particularly noteworthy considering that some findings presented by Nolasco et al. are confirmed by the manuscript. Therefore I believe it would be beneficial to revisit this reference in the discussion section as well. 

3. Line 137: Check the color of the text.

4. Line 378: Please provide more information about this tool (i.e. URL, version)

Comments on the Quality of English Language

I suggest a quick review of the English language used in this document in order to correct any typos.

The manuscript titled "Exploring the Sustainable Valorization of Coffee Silverskin: Extraction of Phenolic Compounds and Proteins for Enzymatic Production of Bioactive Peptide" discusses a study focused on developing innovative extraction methods for phenolic compounds and proteins from Coffee Silverskin (CS), a by-product of the coffee roasting process. Overall, the aims of the paper reflect a comprehensive approach to addressing environmental challenges through scientific innovation, with the potential to make significant contributions to the fields of waste management, sustainable agriculture, and green chemistry. I believe that the paper has enough quality to be published in Foods.

Some observations:

1. The reference by Nolasco et al. is duplicated (1 and 33). Please check accordingly.

2. In some aspects, the article addresses the same points as Nolasco et al. [33], but this reference is scarcely used for an in-depth discussion of the results. This is particularly noteworthy considering that some findings presented by Nolasco et al. are confirmed by the manuscript. Therefore I believe it would be beneficial to revisit this reference in the discussion section as well. 
3. Additionally, I recommend that the authors briefly touch upon the safety aspects of Coffee Silverskin, an area already explored in another work by Nolasco A, Squillante J, Esposito F, Velotto S, Romano R, Aponte M, Giarra A, Toscanesi M, Montella E, Cirillo T in their comprehensive study 'Coffee Silverskin: Chemical and Biological Risk Assessment and Health Profile for Its Potential Use in Functional Foods' (Foods. 2022; 11(18):2834. https://doi.org/10.3390/foods11182834). I believe that acknowledging and citing findings related to safety concerns addressed by these authors could enrich the discussion, offering a balanced view on the utilization of Coffee Silverskin in functional foods. 

4. The authors could also briefly describe other production wastes that have been characterized in recent studies. See also:

  1. Irigoytia, M.B.; Irigoytia, K.; Sosa, N.; de Escalada Pla, M.; Genevois, C. Blueberry by-product as a novel food ingredient: Physicochemical characterization and study of its application in a bakery product. J. Sci. Food Agric. 2022, 102, 4551–4560. 
  2. Nolasco, A.; Squillante, J.; Velotto, S.; D’Auria, G.; Ferranti, P.; Mamone, G.; Errico, M.E.; Avolio, R.; Castaldo, R.; De Luca, L.; et al. Exploring the Untapped Potential of Pine Nut Skin By-Products: A Holistic Characterization and Recycling Approach. Foods 2024, 13, 1044. https://doi.org/10.3390/foods13071044
  3. Ainsa, A.; Marquina, P.L.; Roncalés, P.; Beltrán, J.A.; Calanche M, J.B. Enriched fresh pasta with a sea bass by-product, a novel food: Fatty acid stability and sensory properties throughout shelf life. Foods 2021, 10, 255. 
  4. Kraithong, S.; Issara, U. A strategic review on plant by-product from banana harvesting: A potentially bio-based ingredient for approaching novel food and agro-industry sustainability. J. Saudi Soc. Agric. Sci. 2021, 20, 530–543. 

5. Line 137: Check the color of the text.

6. Line 378: Please provide more information about this tool (i.e. URL, version)

Author Response

Re: [Foods] Manuscript ID: foods-2957069 - Minor Revisions

Manuscript Number: foods-2957069

Respond to the Reviewers

We would like to thank the reviewer for giving valuable comments and suggestions to the manuscript number foods-2957069. We have read your comments carefully and made the correction. The revisions have been made point-by-point according to reviewer’s comments and suggestions. All modifications are highlighted in yellow.

Reviewer 1

Comments and Suggestions for Authors

The manuscript titled "Exploring the Sustainable Valorization of Coffee Silverskin: Extraction of Phenolic Compounds and Proteins for Enzymatic Production of Bioactive Peptide" discusses a study focused on developing innovative extraction methods for phenolic compounds and proteins from Coffee Silverskin (CS), a by-product of the coffee roasting process. Overall, the aims of the paper reflect a comprehensive approach to addressing environmental challenges through scientific innovation, with the potential to make significant contributions to the fields of waste management, sustainable agriculture, and green chemistry.

Response:

            Thank you for your comments and suggestions. Please consider our responses below.

1.1 Comments on the Quality of English Language

I suggest a quick review of the English language used in this document in order to correct any typos.

Response:

            Thank you for your suggestion. We agreed with you. Typos and grammar have been checked through the entire manuscript.

Some observations:

1.2 The reference by Nolasco et al. is duplicated (1 and 33). Please check accordingly.

Response:

            Thank you for your valuable comment. The reference has been checked, and the other references have been checked as well. Please consider Pages 16-19, Lines 513-641.

1.3 In some aspects, the article addresses the same points as Nolasco et al. [33], but this reference is scarcely used for an in-depth discussion of the results. This is particularly noteworthy considering that some findings presented by Nolasco et al. are confirmed by the manuscript. Therefore I believe it would be beneficial to revisit this reference in the discussion section as well.

Response:

            Thank you for your comments and suggestion. We agreed with you. The findings reported by Nolasco et al. (2022) have been used for discussion. Please consider Page 14, Lines 394-395 and Page 16, Lines 482-484.

“Enzymatic hydrolysis by proteases is an interesting process to generate bioactive pep-tides due to the high specificity of the enzymes and mild condition required [1].”

“Moreover, Nolasco et al. (2022) reported that the CS-protein hydrolysate provides ACE inhibitory peptides, antioxidants and DPP IV that possibly uses in nutraceuticals as conventional drugs [1].”

1.4 Additionally, I recommend that the authors briefly touch upon the safety aspects of Coffee Silverskin, an area already explored in another work by Nolasco A, Squillante J, Esposito F, Velotto S, Romano R, Aponte M, Giarra A, Toscanesi M, Montella E, Cirillo T in their comprehensive study 'Coffee Silverskin: Chemical and Biological Risk Assessment and Health Profile for Its Potential Use in Functional Foods' (Foods. 2022; 11(18):2834. https://doi.org/10.3390/foods11182834). I believe that acknowledging and citing findings related to safety concerns addressed by these authors could enrich the discussion, offering a balanced view on the utilization of Coffee Silverskin in functional foods.

Response:

            Thank you for your suggestion. A discussion regarding the safety of coffee silverskin to be used as a functional food has been made. The suggested reference has been used accordingly. Please consider Page 13, Lines 362-365.

            “Moreover, a recent report noted that CS did not show carcinogenic risk. Therefore, CS is safe to be valorized and used as an alternative ingredient for functional foods and pharmaceutical applications [29].”

1.5 The authors could also briefly describe other production wastes that have been characterized in recent studies. See also:

Response:

            Thank you for your suggestions. All suggested references have been used to describe the up-to-date studies about valorization of wastes into value compounds for food applications. The text has been provided in the discussion part (Page 13, Lines 357-361).

            “Previous studies have reported the way to deal with several wastes from different sectors such as blueberry by-product, pine nut skin, sea bass by-product, and plant by-product from banana harvesting [25–28]. After valorization, the value compounds are obtained and can be applied in food application.”

1.6 Line 137: Check the color of the text.

Response:

            Thank you for your comment. The color of the text has been checked accordingly (Page 3, Line 121).

1.7 Line 378: Please provide more information about this tool (i.e. URL, version)

Response:

            Thank you for your comment. The URL of Peptide Ranker has been provided (Page 12, Line 342). In addition, we have checked the URL of other webservers used for in silico analysis. Please consider Page 13, Line 349-350.

Reviewer 2 Report

Comments and Suggestions for Authors

The comments for the authors:

1. I suggest removing the term "Exploring the" from the title.

2. The authors should avoid using first person plural throughout the text.

3. The abbreviations should be defined just the first time they are mentioned in the text and used as such.

4. Particle size and moisture content of dried sample material are missing in the subsection 2.1.

5. Why did the authors use hydrothermal treatment, especially temperatures of 110-130oC when it is known that phenolic compounds are thermosensitive? There are other extraction techniques such as ultrasound-assisted extraction that is more efficient in terms of yield and costs.

6. Calibration curve and correlation factor for TPC is missing.

7. The authors stated that they optimized the process. In this sense, they should use a specific extraction optimization model/program. Has it been done?

8. Statistical analysis should be provided in Figure 1 and 2. The quality of Figure 1 should be improved.

9. The authors mentioned that the HT-extracted CS liquid and solid fractions were used for further analysis. However, it is not stated at which extraction conditions these fractions were obtained.

10. Table 2. The meaning of different letter within the columns should be explained in the caption.

11. "The coffee silverskin is composed of various variable substances such as phenolic compound, 560 protein, and fibers." This sentence is not relevant for Conclusions. 

Author Response

Re: [Foods] Manuscript ID: foods-2957069 - Minor Revisions

Manuscript Number: foods-2957069

Respond to the Reviewers

We would like to thank the reviewer for giving valuable comments and suggestions to the manuscript number foods-2957069. We have read your comments carefully and made the correction. The revisions have been made point-by-point according to reviewer’s comments and suggestions. All modifications are highlighted in yellow.

Reviewer 2

Comments and Suggestions for Authors

2.1 I suggest removing the term "Exploring the" from the title.

Response:

            Thank you for your suggestion. We agreed with you, and the term “Exploring” has been moved from the title. The revised title is “Sustainable Valorization of Coffee Silverskin: Extraction of Phenolic Compounds and Proteins for Enzymatic Production of Bioactive Peptide”.

2.2 The authors should avoid using first person plural throughout the text.

Response:

            Thank you for your comment. We agree with you. The use of first-person plural has been checked in the whole manuscript. Please consider Page 1, Lines 15-17.

“The conditions for hydrothermal (HT) extraction of phenolic compounds from CS were optimized by varying CS loading (2.5−10%, w/v), temperature (110−130°C), and time (5−30 min) using one-factor-at-a-time (OFAT) approach.”

2.3 The abbreviations should be defined just the first time they are mentioned in the text and used as such.

Response:

            Thank you for your comment. The use of abbreviations has been checked. Here, you will find the revisions:

  • Page 1, Line 34: coffee silverskin (CS) [1]. Currently, the majority of CS
  • Page 2, Lines 66-67: a protease from halodurans SE5 (protease_SE5)
  • Page 2, Line 76 and Line 86: a one-factor-at-a-time (OFAT) approach and OFAT
  • Page 3, Line 100 and Line 107: the terms for conventional alkaline extraction (CAE), Ultrasound-assisted alkaline extraction (UAE), and microwave-assisted alkaline extraction (MAE)
  • Page 4, Line 137: The molecular weight (MW) distribution of low MW
  • Page 4 Line 158, 161, and 168: the terms for antioxidant activity assays

2.4 Particle size and moisture content of dried sample material are missing in the subsection 2.1.

Response:

            Thank you for your comment. The missing details have been provided. Please consider Page 2, Lines 74-75.

“The particle size and moisture content of CS were 100 mesh and 5.60±0.23% (w/w), respectively.”

2.5 Why did the authors use hydrothermal treatment, especially temperatures of 110-130°C when it is known that phenolic compounds are thermosensitive? There are other extraction techniques such as ultrasound-assisted extraction that is more efficient in terms of yield and costs

Response:

            Thank you for your comments and suggestion. According to the findings of this study, the optimal temperature for extraction of phenolic compounds from coffee silver skin was 125°C. Further increasing temperature can cause the degradation of phenolic compounds in the extracts, and the antioxidant activity was consequently decreased. The results agreed with the previous study where phenolic compounds were extracted from coffee silverskin. To clarify this point, more discussion has been made (Page 14, Lines 378-382).

“The result was in accordance with the previous study in which extraction temperature <200°C, did not negatively affect phenolic compounds and antioxidant activity of CS extracts. However, TPC and antioxidant activity decreased when extraction temperature was above 200°C [30].”

Additionally, hydrothermal treatment is a chemical free process, so the extracts can be used in food and pharmaceuticals applications without any additional step.

2.6 Calibration curve and correlation factor for TPC is missing.

Response:

            Thank you for your comment. The calibration curve and equation with R2 for calculation of total phenolic content (TPC) have been provided (Page 4, Lines 180-182).

2.7 The authors stated that they optimized the process. In this sense, they should use a specific extraction optimization model/program. Has it been done?

Response:

            Thank you for your comments. In this study, the effect of individual factors, CS loading, temperature, and extraction time were studied by one-factor-at-a-time (OFAT) approach. The main goal of this study was to observe the impact of each factor on the extraction efficiency as well as the preliminary range that can be used for further studies. For further study, we have a plan to optimize the extraction conditions by using a statistical design by either central composite design (CCD) or Box–Behnken design (BBD). By doing this, we will see the interaction between factors, and get the model for prediction.

2.8 Statistical analysis should be provided in Figure 1 and 2. The quality of Figure 1 should be improved.

Response:

            Thank you for your comments. The statistical analysis was applied to the experimental data in Figure 1 and 2. In addition, the quality of Figure 1 was improved by adjusting resolution. Please consider the modified version of Figure 1 and Figure 2.

2.9 The authors mentioned that the HT-extracted CS liquid and solid fractions were used for further analysis. However, it is not stated at which extraction conditions these fractions were obtained.

Response:

            Thank you for your comments. Initially, the optimal conditions for extraction of phenolic compound from coffee silverskin (CS) were optimized by one-factor-at-a-time (OFAT) approach. Then, the optimal condition was used for extracting phenolic compounds, which is the extract or liquid fraction (HT-extracted CS liquid fraction). The liquid fraction was further analyzed for phenolic compounds by LC-QTOF-MS.

After the extraction, another remaining part is the solid residue (HT-extracted CS solid fraction), and this fraction was then used for protein extraction to produce bioactive peptide. Overall, the extraction conditions are the condition optimized by OFAT study.

            To clarify this point, the extraction conditions have been stated. Please consider Page 2, Lines 85-86 and Page 6, Lines 211-213.

2.10 Table 2. The meaning of different letter within the columns should be explained in the caption.

Response:

            Thank you for your suggestion. We agree with you. The meaning of different letters within columns are defined by the footnote (Page 8, Lines 272-273). In addition, the sentence is provided to describe different letters annotated by statistical analysis for Figure 1, 2, and 5.

2.11 "The coffee silverskin is composed of various variable substances such as phenolic compound, 560 protein, and fibers." This sentence is not relevant for Conclusions.

Response:

            Thank you for your suggestion. We agree with you. The term “fibers” has been removed from the text.
